# YidC from *Escherichia coli* Forms an Ion-Conducting Pore upon Activation by Ribosomes

**DOI:** 10.3390/biom13121774

**Published:** 2023-12-11

**Authors:** Denis G. Knyazev, Lukas Winter, Andreas Vogt, Sandra Posch, Yavuz Öztürk, Christine Siligan, Nikolaus Goessweiner-Mohr, Nora Hagleitner-Ertugrul, Hans-Georg Koch, Peter Pohl

**Affiliations:** 1Institute of Biophysics, Johannes Kepler University Linz, Gruberstrasse 40, A-4020 Linz, Austria; denis.knyazev@jku.at (D.G.K.); winter.lukas22@gmail.com (L.W.); sandra.posch@jku.at (S.P.); christine.siligan@jku.at (C.S.); nikolaus.goessweiner-mohr@jku.at (N.G.-M.); nora.hagleitner-ertugrul@jku.at (N.H.-E.); 2Institute of Biochemistry and Molecular Biology, ZBMZ, Faculty of Medicine, Albert Ludwig University of Freiburg, 79104 Freiburg, Germanyyavuz.oeztuerk@biochemie.uni-freiburg.de (Y.Ö.); hans-georg.koch@biochemie.uni-freiburg.de (H.-G.K.); 3Spemann-Graduate School of Biology and Medicine (SGBM), Albert Ludwig University of Freiburg, 79104 Freiburg, Germany; 4Faculty of Biology, Albert Ludwig University of Freiburg, 79104 Freiburg, Germany

**Keywords:** protein translocation, fluorescence correlation spectroscopy, electrophysiology, single dye tracing

## Abstract

The universally conserved protein YidC aids in the insertion and folding of transmembrane polypeptides. Supposedly, a charged arginine faces its hydrophobic lipid core, facilitating polypeptide sliding along YidC’s surface. How the membrane barrier to other molecules may be maintained is unclear. Here, we show that the purified and reconstituted *E. coli* YidC forms an ion-conducting transmembrane pore upon ribosome or ribosome-nascent chain complex (RNC) binding. In contrast to monomeric YidC structures, an AlphaFold parallel YidC dimer model harbors a pore. Experimental evidence for a dimeric assembly comes from our BN-PAGE analysis of native vesicles, fluorescence correlation spectroscopy studies, single-molecule fluorescence photobleaching observations, and crosslinking experiments. In the dimeric model, the conserved arginine and other residues interacting with nascent chains point into the putative pore. This result suggests the possibility of a YidC-assisted insertion mode alternative to the insertase mechanism.

## 1. Introduction

Protein insertion into the cytoplasmic membrane is an essential process in bacterial physiology. YidC is a member of the Oxa1 family of protein insertases [1] and is involved in inserting proteins into the bacterial cytoplasmic membrane. Multiple pieces of evidence demonstrate that YidC can act in two modes: independently [2,3,4] and assisting the major bacterial protein-conducting channel SecYEG [5,6,7,8]. Two partially redundant membrane protein integration systems, YidC and SecY, may be required because the number of SecYEG complexes in the bacterial membrane is relatively small [9]. A second insertion site for less complex membrane proteins would ease the transport load of the SecYEG translocon. In support of this, the SRP pathway can deliver some multi-spanning membrane proteins to either SecYEG or YidC, suggesting that some membrane proteins’ insertion pathway is promiscuous [4].

In the SecYEG assisting mode, YidC is a part of the SecYEG pore [10], where it seems to exist as a monomer that binds to the lateral gate of SecYEG [11], most probably via the intercalation of the N-terminal transmembrane domain [12,13]. In this mode, YidC is theorized to enhance the release of transmembrane helices from the SecY channel, facilitate their subsequent folding [6,7,8], and even promote the assembly of multi-subunit membrane protein complexes [14]. Whether YidC can also form pores for protein insertion when acting independently of the SecYEG translocon is unknown. Crystallographic data suggest that the functional YidC unit is a monomer, as was demonstrated for the truncated and full versions of YidC from *Thermotoga maritima* in detergent micelles (TmYidC) [13,15], and for *Bacillus halodurans* YidC2 (BhYidC) [16] and *E. coli* YidC (EcYidC) [17] in the lipidic cubic phase. The insertion mechanism inspired by the structural data includes sliding a substrate’s hydrophilic and negatively charged N-terminus through the hydrophobic surface of YidC’s water-filled groove, which contains a conserved arginine residue (R366 in EcYidC) [18,19]. Although the arginine has been shown to be important, it does not appear to be essential in E. coli YidC unless the hydrophilicity of the groove is reduced [19].

Exposing a charge to the hydrophobic membrane interior—as proposed for R366—is a rare feature for a transmembrane protein. The associated energetic penalties for (i) ion dehydration (Born energy) and (ii) placing a positive charge into an environment already possessing a positive potential (membrane dipole potential) [20] often force the charged side chains to point away from the hydrophobic lipid core. The voltage sensor of voltage-gated channels may serve as an example [21,22]. Likewise, establishing water-filled grooves is energetically favorable inside the protein—like in potassium or sodium channels [23]—but much less so at the protein–lipid interface. It is also unclear how YidC’s hydrophilic groove ensures specificity for polypeptides and what prevents other ions from being drawn into the water-filled cavity by the positive charge in the center [24].

Polypeptide insertion into the membrane by translocases usually occurs from inside a water-filled pore [25,26]. Recent bioinformatic analyses indicated that the SecY protein channel originated from a dimeric YidC-like protein [27], which suggests that YidC might also function as a pore-forming dimer. The EcYidC contains a large 320 amino acid-long periplasmic loop (P1) between the first two membrane helices, which might facilitate dimerization. This function is supported by the crystal structure of the periplasmic domain of EcYidC, which exhibits a dimer [28,29], and by experiments on native *E. coli* membranes [30]. The dimerization interface on the bacterial YidC may be on TM3 [31] or TM5 [27]. For other Oxa1 family members, such as the eukaryotic EMC3 and GET1, dimerization depends on TM5 [32,33,34].

Reconstituted planar lipid bilayers proved helpful in characterizing SecYEG’s pore and the elements required to maintain the membrane barrier to small molecules [35]. We then used the same model to test the possibility of pore formation by YidC. We observed channel activity in YidC-containing bilayers in the presence of ribosomes or ribosome-nascent chains (RNCs) of the natural YidC substrate FoC. We also assessed the oligomeric state of the reconstituted YidC in vesicles, IMVs, and planar bilayers with fluorescence correlation spectroscopy (FCS), BN-PAGE, in vivo site-directed crosslinking, and single-molecule microscopy. Based on a new structural model generated by AlphaFold, we conclude that the YidC dimer can form an ion-conductive pore lined by residues known to interact with nascent chains.

## 2. Materials and Methods

### 2.1. Cell Culture

In vivo experiments were conducted using *E. coli* strain BL21 (Merck, Darmstadt, Germany) and its plasmids, pEVol [36], and pTrc99a-SecYEG_His_-YidC [12,37]. Cells were grown in LB medium at either 30 °C or 37 °C.

### 2.2. The Cloning of pTrc99a-YidCΔC and Construction of a Single-Cysteine YidC Derivative

pTrc99a-YidCΔC was cloned from pTrc99a-YidC [4] by inverse PCR deleting the last 13 amino acids of YidC using the following primer pair:

YidCdeltaCRev: (5′-GCTGATTTACCGTGGTCTG-3′); YidCdeltaCFwd: (5′-TGATTCGGTGAGTTT TCG-3′). For the labeling of YidC with ATTO488, a single cysteine mutation was introduced via site-directed mutagenesis at position 269 (YidC_D269C_fwd: 5′-CCCGCATAACTGCGGTACCAACAACTTC-3′; YidC_D269C_rev: 5′-GAAGTTGTTGGTACCGCAGTTATGCGGG-3′) in a cysteine-free YidC (C423S) generated via site-directed mutagenesis from pTrc99a-YidC [4] (YidC_C423S_fwd: (5′-CCGCTGGGCGGCTCCTTCCCGCTG-3′; YidC_C423S_rev: G CAG CGG GAA GGA GCC GCC CAG CGG). The D269C mutant has been previously used for the fluorescent labeling of YidC [38].

### 2.3. The Purification and Reconstitution of YidC into Vesicles

pTrc99a-SecY_His_EG-YidC was used as a template for the generation of the YidC-pBpa derivatives YidC (D399pBpa) and YidC (E544pBpa) [12,37]. The following *E. coli* strains and plasmids were used: DH5α [39], BL21 (Merck, Darmstadt, Germany), pEVol [36], and pTrc99a-SecYEG_His_-YidC [12]. Cells were grown in LB medium at either 30 °C or 37 °C.

YidC and YidCΔC were purified from pTrc99a-YidC- [4] and pTrc99a-YidCΔC-expressing BL21 cells using a Ni-NTA FF crude column (GE Healthcare, Chicago, IL, USA) on an ÄKTA chromatography system. The equilibration/wash buffer contained 50 mM Tris-HCl pH 7.5, 300 mM NaCl, 5 mM MgCl_2_, 20 mM imidazole, 0.03% DDM (Affymetrix Antrace, Maumee, OH, USA), 10% glycerol, and His-tagged proteins were eluted with a linear gradient from 20 to 500 mM imidazole. Dried *E. coli* polar phospholipids (Avanti Polar Lipids, Alabaster, AL, USA) were rehydrated in buffer (50 mM Tea-OAc pH. 7.5, 50 mM DTT) to a final concentration of 100 mg/mL and sonicated. Proteoliposomes were prepared by mixing lipids (final concentrations of 0.1 mg/mL) and 0.85% (*w*/*v*) n-octyl-β-D-glycoside with 1.5 µM purified and DDM-solubilized proteins and incubating them for 20 min at 4 °C. Samples were then dialyzed (Spectrapor membrane tubing, 6–8 kDa) against 50 mM TeaOAc, pH 7.5, 1 mM DTT. The proteoliposomes were pelleted (1 h, 210,000× *g*) and resuspended in 50 mM TeaOAc, pH 7.5, 1 mM DTT to a final protein concentration of 5 µM. The YidC-to-lipid ratio was 1:100, m%. The reconstituted YidC is functionally active, as indicated by its ability to insert different membrane proteins like MtlA or TatC [4].

### 2.4. The Labeling of YidC

YidC (D269C, C423S) was labeled with ATTO488 on the periplasmic loop before reconstitution. EcYidC’s loop is almost as big as BhYidC’s. Therefore, the purified protein was incubated with 100 μM TCEP for 5 min on ice before incubating with a ten-fold molar excess of ATTO-488-maleimide (Sigma-Aldrich, Vienna, Austria) for 2 h on ice. Excess dye was removed by desalting via PD10 columns (GE Healthcare) before reconstitution.

### 2.5. BN-PAGE Analysis

Purified IMVs (100 μg protein) of wild-type *E. coli* TY0 cells or the conditional YidC depletion strain JS7131 [40] were dissolved in a buffer containing 50 mM imidazole/HCl, pH 7.0, 5 mM 6-aminocaproic acid, 50 mM NaCl, solubilized with 1% final concentration n-Dodecyl β-D-maltoside (Roche, Mannheim, Germany), and incubated for 5 min at 25 °C. Non-solubilized material was pelleted via centrifugation for 30 min at 45,000 rpm, 4 °C (TLA-45 rotor, Beckmann Coulter, Krefeld, Germany). The solubilized proteins were separated on 4–15% BN-gels and analyzed via immune detection using polyclonal α-YidC antibodies. Dissociation of YidC dimers was induced by treatment with 0.1% SDS for 10 min at 56 °C before loading on BN-PAGE.

### 2.6. In Vivo pBpa Crosslinking

BL21 cells containing the pEVol plasmid, together with YidC-pBpa variants in pTrc99a plasmids, were cultured overnight in LB medium at 37 °C. In total, 10 mL of the overnight culture was further used for inoculation of the 1 L LB medium containing 0.5 mM pBpa (in NaOH), 50 μg/μL of ampicillin, and 35 μg/μL of chloramphenicol. The cultures were further incubated at 37 °C until they reached the early exponential growth phase (OD_600_ = 0.5–0.8), and induced with 0.02% L-arabinose (for pEVol plasmid) and 0.5 mM isopropyl 1-thio-β-D-galactopyranoside (IPTG, for pTrc99a plasmids). After induction, the cultures were grown for 2 h at 37 °C, cooled down on ice for 10 min and harvested via centrifugation at 5000 rpm in a JLA 9.1000 rotor for 10 min. The cell pellets were resuspended in 10 mL of PBS buffer (137 mM NaCl, 2.7 mM KCl, 10 mM Na_2_HPO_4_, and 1.76 mM KH_2_PO_4_) and divided in two multi-well plates. One plate was exposed to UV light on ice for 20 min (UV chamber: BLX-365, from Vilber Lourmat, Eberhardzell, Germany) while the other plate was kept in the dark. After UV irradiation, the cell suspension was transferred to 50 mL Falcon tubes and cells were collected via centrifugation at 5000 rpm for 10 min in a table-top centrifuge. Each cell pellet was resuspended in 10 mL of resuspension buffer (50 mM Tris/HCl pH 7.5, 300 mM NaCl, 10 mM Mg(Ac)_2_) for subsequent YidC purification. Next, the samples were lysed by French pressing and the cell debris was removed via centrifugation at 15,800 rpm for 30 min in a SS34 rotor. The supernatant was further centrifuged at 45,000 rpm for 1.5 h in a TLA 50.2 rotor for membrane sedimentation. Next, the membrane pellets were solubilized in 1% (*w*/*v*) n-dodecyl β-D-maltoside (DDM) dissolved in resuspension buffer supplemented with 10% (*w*/*v*) glycerol for 1 h at 4 °C. YidC was purified via metal affinity chromatography using TALON^®^ Metal affinity resin (Clontech, Mountain View, CA, USA). The samples were incubated for 1 h at 4 °C with 1 mL/1 L LB culture of pre-equilibrated TALON^®^ Metal affinity resin, washed five times with 10 mL/sample of washing buffer (40 mM Imidazol, 10% glycerol (*w*/*v*), 50 mM Tris/HCl, pH 7.5, 300 mM NaCl, 10 mM Mg(Ac)_2_, and 0.03% DDM) and eluted four times in a total volume of 2 mL elution buffer (200 mM Imidazole, 10% glycerol (*w*/*v*), 50 mM Tris/HCl pH 7.5, 300 mM NaCl, 10 mM Mg(Ac)_2_, and 0.03% DDM). The samples were then precipitated with 1 volume of 10% trichloracetic acid (TCA), denatured at 56 °C for 10 min in 35 μL of TCA loading dye (prepared by mixing one part of Solution III (1 M Dithiothreitol) with 4 parts of Solution II (6.33% SDS (*w*/*v*), 0.083 M Tris-Base, 30% glycerol, and 0.053% Bromphenol blue) and 5 parts of Solution I (0.2 M Tris, 0.02 M EDTA pH 8)) and analyzed on SDS Page using Western blotting.

### 2.7. CyoA Leader Peptide

We tested the signal peptide of the precursor form of subunit A of the cytochrome o oxidase (sequence: MRLRKYNKSLGWLSLFAGTVLLSG). It was synthesized and purified to 98% purity by Peptide 2.0 Inc. (Chantilly, VA, USA).

### 2.8. Reconstitution of YidC into Planar Bilayers

Planar bilayers were formed in a Teflon chamber with two compartments separated by a Teflon septum. Pure lipid vesicles from E.coli polar lipid extract (Avanti Polar Lipids) were added to the two compartments containing 50 mM K-HEPES and 150 mM KCl (pH = 7.5). The final lipid concentration was between 1 and 2 mg/mL. Planar bilayers were then folded by raising the level of these two aqueous solutions over the dividing aperture in the Teflon septum, thereby combining the two lipid monolayers on top [41]. Then, 30 to 60 min of control current recordings ensured the absence of lipid channels.

We used two different approaches for the subsequent protein reconstitution: (i) We lowered the level of the aqueous solutions below the aperture, added YidC-containing proteoliposomes, and raised the buffer levels above the aperture after an incubation time of ~1 h [35,42]. (ii) We added proteoliposomes and ribosomes to one side of the intact membrane. Increasing the osmolarity in that compartment by adding 300 mM KCl resulted in YidC insertion into the planar membrane by vesicle fusion [43]. Ribosomes were added to a final concentration of 0.5–1 mg/mL. For experiments with ribosome-nascent chain complexes, RNCs were added to a final concentration of 0.5–1 µg/mL.

### 2.9. Single Ion Channel Measurements

Ag/AgCl reference electrodes were immersed into the buffer solutions on both sides of the lipid bilayer. The command electrode of the patch clamp amplifier (model EPC9, HEKA electronics, Lambrecht, Germany) was immersed into the cis compartment, and the ground electrode into the trans compartment. The recording filter for the transmembrane current was a 4-pole Bessel with a −3 dB corner frequency of 0.1 kHz. The raw data we obtained were analyzed using the TAC software 4.0 package (Bruxton Corporation, Seattle, WA, USA). Gaussian filters of 12 Hz were applied to reduce noise.

### 2.10. Ribosome Expression and Purification

Tetra-(His)_6_-tagged ribosomes from the *E. coli* JE28 strain were purified as described previously [44]. An overnight culture of *E. coli* JE28 was used to inoculate 1 L LB medium supplemented with 50 µg/mL kanamycin. The cells were grown to an OD_600_ of 1.0 at 37 °C. After that, the culture was kept at room temperature for 1 h before shifting it to 4 °C for another hour to produce run-off ribosomes. The cells were harvested via centrifugation at 4000 rpm for 30 min. For purification, the cell pellet was resuspended in lysis buffer (20 mM Tris-HCl pH 7.6, 10 mM MgCl_2_, 150 mM KCl, 30 mM NH_4_Cl) with 0.5 mg/mL lysozyme and 10 µg/mL DNAse I, and lysed using a BeadBeater (BioSpec, Bartlesville, OK). The lysate was then clarified via centrifugation and passed over a Ni^2+^-chelating column. The ribosomes were eluted with 150 mM imidazole and then dialyzed overnight against lysis buffer. The ribosomes were then pelletized via ultracentrifugation and resuspended in 500 mM NH_4_Cl, 50 mM Tris-acetate, and 25 mM Mg-acetate, resulting in a final concentration of 10–20 mg/mL. The pH was adjusted to 7.2. All buffers were supplemented with complete protease inhibitor cocktail (Roche).

### 2.11. Labeling of Ribosomes after Purification

Nonspecific labeling of ribosomes was obtained as described previously [45]. For this, the purified ribosome complex was incubated with Atto-633-NHS ester in 20 mM Tris-HCl, 10 mM MgCl_2_, 150 mM KCl, and 5 mM NH_4_Cl for 30 min at 37 °C. Labeled ribosomes were then separated from unbound dye via ultracentrifugation for 3 h at 40,000 rpm and 4 °C, using a Beckman Coulter ultracentrifuge (Rotor Type 90 Ti, Beckmann Coulter, Krefeld, Germany). The pellet containing the ribosomes was washed four times with 140 µL 150 mM KCl and 50 mM K-HEPES and then resuspended in the same buffer. The pH was always 7.5.

### 2.12. Single-Molecule Microscopy Measurements

Single-molecule fluorescence microscopy measurements were performed on the inverse microscope Olympus IX83 equipped with Toptica iBeam-Smart lasers for 640 and 488 nm and an Andor iXon3 EMCCD camera. The YidC vesicles were mixed with empty liposomes in a ratio of 1:1000. The empty liposomes were prepared from DOPE/DOPG 7/3 mol.% via an extrusion technique using polycarbonate membranes with a pore diameter of 100 nm (Avestin, Mannheim, Germany). Synthetic lipids DOPE and DOPG were used instead of *E. coli* polar lipid extract to avoid lipid autofluorescence.

### 2.13. Fluorescence Correlation Spectroscopy (FCS)

Fluorescence correlation spectroscopy (FCS) detected the ribosome binding to proteoliposomes and determined the number of proteins per vesicle. In brief, the average residence time *τ_D_* and the number of labeled ribosomes in the confocal volume were derived from the autocorrelation function G(τ) of the temporal fluorescence signal, which was acquired using a commercial laser scanning microscope equipped with avalanche diodes (LSM 510 META ConfoCor 3, Carl Zeiss, Jena, Germany) and a 40× water immersion objective. The diffusion coefficient *D* was determined as ω^2^/4*τ_D_*, where ω is the radius of the focal plane. The residence time *τ_D_* was obtained from the standard model for one-component free 3D diffusion [46]:$$G(\tau )=1+\frac{1}{n(1+\frac{\tau }{\tau_{D}})}$$
where *n* is the number of fluorescent particles in the confocal volume.

To determine the number of proteins per vesicle [47], we first counted the number of proteoliposomes in the confocal volume using the ATTO-488-label on YidC. Next, we compared the number of particles per confocal volume before and after the solubilization of YidC vesicles with 3% octyl glucoside (OG). The detergent concentration was chosen to be well above the critical micelle concentration and therefore solubilize the liposomes. The increase in the number of particles then provides a good estimation of the proteins per vesicle. A subsequent addition of 1.1% SDS was performed to examine a possible oligomeric state of the YidC in OG micelles.

### 2.14. Ribosome Nascent Chains

RNCs of F_0_c were prepared in vivo from *E. coli* KC6(DE3) harboring pBAD-F0c(1–46) (a gift from E.O. van der Sluis and Roland Beckmann [48]). To generate RNCs in vivo, cells were grown at 37 °C to an OD600 of 0.5 and induced for 1 h with 0.2% arabinose. Subsequently, the cultures were cooled down on ice, then harvested via centrifugation for 10 min in a SLC 6000 rotor at 5500 rpm and 4 °C. The cell pellet was resuspended in 1.5 mL RNC buffer per gram of wet cell pellet (50 mM Tea-acetate, pH 7.5; 150 mM KOAc; 10 mM Mg(OAc)_2_; 1 mM tryptophan; 250 mM sucrose) in the presence of cOmplete EDTA-free protease inhibitor. Cells were lysed by passing them through a French press at 8000 psi. A final concentration of 0.1% DDM was added. Cell debris was removed at 16,000 rpm and 4 °C for 20 min in a SS34 rotor. To separate RNCs from membranes and the cytosolic fraction, the lysate was overlaid on a sucrose cushion (RNC buffer with 750 mM sucrose) in a 1 to 2 ratio and centrifuged in a Ti50.2 rotor for 17 h at 24,000 rpm and 4 °C. The pellet was then resuspended in 4 mL RNC buffer supplemented with 0.1% DDM by vigorous shaking. After complete resuspension, it was applied onto pre-equilibrated TALON material (2 mL slurry washed once with water, twice with RNC buffer with 0.1% DDM, and once with RNC buffer supplemented with 0.1% DDM and 10 μg/mL yeast in a propylene column). After 1 h of binding at 4 °C, the TALON material was washed five times with 10 column volumes of RNC buffer lacking tryptophan. RNCs were eluted by 6 consecutive elution steps of 1 mL RNC elution buffer (50 mM Tea-acetate, pH 7.5; 150 mM KOAc; 10 mM Mg(OAc)2; 150 mM imidazole; 250 mM sucrose; complete EDTA-free protease inhibitor). The eluted fractions were pooled, and RNCs were collected for 2 h in a TLA100.3 rotor at 40,000 rpm and 4 °C. The ribosomal pellet was resuspended in 300 μL buffer (50 mM Tea-acetate, pH 7.5; 150 mM KOAc; 10 mM Mg(OAc)_2_), aliquoted, flash frozen in liquid N_2_, and stored at −80 °C.

### 2.15. Calculation of Channel Ion Selectivity

The potential, $\psi_{r}$, at which the ionic current through the channel equals zero was taken from either single-channel analysis or current–voltage ramp recordings. From this so-called reversal potential, the anion ($Cl^{-}$) to cation ($K^{+}$) permeability ratio ($\frac{P_{Cl}}{P_{K}}$) was calculated according to Goldman’s equation for bi-ionic potentials, with ion concentrations $K_{cis}^{+}$, $Cl_{cis}^{-}$ and $K_{trans}^{+}$, $Cl_{trans}^{-}$ in *cis* and *trans* compartments:$$\psi_{r}=\frac{RT}{F}ln(\frac{K_{cis}^{+}+\frac{P_{Cl}}{P_{K}}Cl_{trans}^{-}}{K_{trans}^{+}+\frac{P_{Cl}}{P_{K}}Cl_{cis}^{-}})$$

For potassium and chloride concentrations in the two compartments, we account for the osmotic water flow within the unstirred water layers near the membrane, which concentrates the solution on the hypoosmotic side and dilutes it on the hyperosmotic side of the membrane. For such membranes, this effect does not usually exceed 10%. Therefore we assumed the bulk KCl gradient of 450 to 150 mM corresponded to a 230 mM gradient adjacent to the lipid bilayer.

### 2.16. Estimation of the Pore Diameter

A rough estimation of the pore diameter can be obtained from the single-channel conductance using
$$\frac{1}{g}=\left(l+\frac{\pi d}{4}\right)\frac{4}{\sigma \pi d^{2}}\;$$
where d, l and σ are channel diameter, channel length, and conductivity of the solution, respectively [35]. σ = 3.7 S/m was taken as an average from the conductivities of the two compartments; the channel length was assumed to be l = 3 nm.

### 2.17. Modeling of YidC Dimers with AlphaFold

For modeling likely YidC dimer conformations, the *E. coli* YidC sequence (UniProt: P25714) was used as the query_sequence in ColabFold (version 1.5.2) [49] for GoogleColab, which utilizes AlphaFold2 [50] and AlphaFold2-multimer [51], as well as MMseqs2 [52] and HHsearch [53], for alignment/template searches. The settings were largely kept as default. *AlphaFold2_multimer_v3* was used as the AlphaFold *model_type*. The resulting models were analyzed for their plausibility, and model visualizations were generated in PyMOL.

## 3. Results

### 3.1. Purified and Reconstituted YidC Displays Ion Channel Activity in the Presence of Its Substrate FoC or Empty Ribosomes

Liposomes with reconstituted YidC were fused to a pre-formed lipid planar bilayer in the presence of FoC-RNCs, consisting of the transmembrane domain of FoC (residues 1–46) followed by an HA-tag and a TnaC stalling sequence [48]. Vesicles were added from the hyper-osmotic side. The vesicle fusion was triggered by an osmotic gradient across the planar bilayer, which caused a destabilization of the contact area between the proteoliposomes and the planar bilayers. The destabilizing effect is due to osmotic water flux directed from the vesicle towards the hypotonic side [43]. Analysis of the single-channel recordings (Figure 1A) revealed a single-channel conductance of 459 ± 20 pS. According to previous reports, SecYEG-RNC complexes and SecYEG-YidC-RNC complexes exhibit similar conductance values (±15%). This observation suggests comparable pore sizes. The size was deemed necessary for the two earlier reported pores (SecYEG, SecYEG-YidC) to accommodate the substrate.

The YidC channel exhibits a reversal potential of −9 mV (Figure 1B). According to the Goldman–Hodgkin–Katz equation, it indicates a less than twofold preference for anions over cations (see Section 2.15). We previously observed this preference for the SecYEG translocon [54]. The modest anion selectivity makes only a very minor contribution to maintaining the proton motive force during protein translocation.

The addition of (i) FoC-RNCs to protein-free bilayers or (ii) YidC vesicles to the hypertonic side without FoC-RNC induced no channel activity (Figure 1D). Hence, the observed channel activity (Figure 1A–C) was due to the complex of YidC and FoC-RNC.

Even though the ability of YidC to form conductive pores in the presence of the SecYEG translocon and RNCs was reported previously [10], we are unaware of any reports about the water or ion conductance of substrate-activated YidC in the absence of SecY. Yet, the homologous yeast Oxa1 forms pores capable of accommodating a translocating protein segment [55].

Since SecYEG channels open upon ribosome binding [56,57], we tested whether YidC possesses the same capability. Instead of FoC-RNC (Figure 1), we added non-translating ribosomes and YidC-containing proteoliposomes to the hypertonic side. Subjecting the induced channel activity (Figure 2A) to a histogram analysis (Figure 2B) revealed a unitary channel conductivity g = 436 ± 20 pS, (Figure 2C) close to the 459 ± 20 pS of the FoC-RNC-YidC complex (Figure 1). However, the RNC concentration needed for observing the channel activity was two to three orders of magnitude lower than that of the non-translating ribosomes. The reversal potential U_rev_ = −5 mV was even smaller than in the presence of an RNC (Figure 1), indicating an even smaller preference for anions in the presence of empty ribosomes as compared to FoC-RNC.

Adding the YidC vesicles in the absence of ribosomes (Figure 2D, lower trace) or ribosomes in the absence of YidC vesicles (Figure 2D, upper trace) did not lead to channel activity. The former proves that YidC is electrically silent without a binding partner, and the latter proves that the purified ribosomes did not contain any channel-forming contaminants.

The only other channel which the ribosomes could activate is SecYEG. Its co-purification together with YidC is unlikely, as confirmed by Western Blot analysis (Appendix A). Records with a high number of YidC channels in a single membrane (Appendix A) also prove that the channel-forming entity cannot be due to a minor contaminant in the purified sample.

The ligand-free YidC remains electrically silent even when exposed to a transmembrane voltage (Appendix A). Instead of using vesicle fusion to insert the protein into the planar bilayer, we folded solvent-depleted planar bilayers from the monolayers that form on top of proteoliposome suspensions [35]. These monolayers also contain membrane proteins [58]. The thus reconstituted YidC was functionally intact, as demonstrated by ribosome-induced channel formation (Appendix A).

Next, we tested whether signal peptides may activate YidC without ribosomes. The membrane insertion of phage proteins (PF3, M13) that are too short to be co-translationally targeted to YidC [2,59] supports this hypothesis. Yet the signal peptide of subunit A of cytochrome o oxidase did not activate YidC. We did not observe channel activity unless we added ribosomes. Notably, cyoA insertion requires both SecY and YidC [60,61].

### 3.2. YidC with C-Terminal Deletion Retains Ribosome Binding Activity

The C-terminus is part of the ribosome-binding site in *E. coli* YidC [4]. Similarly, the much longer C-terminus of the yeast YidC homolog Oxa1 is also known to be involved in ribosome binding [62]. For electrophysiological experiments with YidC’s C-terminal-deletion mutant (YidCΔC), we folded the bilayers from YidC-containing monolayers. The addition of purified empty *E.coli* ribosomes led to the formation of ion-conducting channels of a conductance g = 392 ± 30 pS (Figure 3A–C), which is close to that observed for the complex of empty ribosomes with the full-sized YidC. In line with our result is the observation that deleting the C-terminus reduces ribosome affinity (K_d_ ~ 1 µM) by only approx. 75% [38]. The residual affinity is large enough to result in a significant binding probability.

To confirm this observation, we non-specifically labeled the His-tagged ribosomes with an ATTO-633-NHS ester [45] and measured their diffusion via FCS in solution, in the presence of empty vesicles or YidC proteoliposomes. Upon proteoliposome addition, 40% of the ribosomes showed an increased residence time (Figure 3D), which indicates the binding of the 70S subunit to the YidC vesicles.

Our data show that ribosomes can interact with YidC even in the absence of its C-terminus, confirming previous studies [38,63], which concluded that motifs, in addition to the C-terminus, are also involved in binding to ribosomes. The His-tag introduced into all our YidC constructs for affinity purification might facilitate the residual binding of ribosomes to YidCΔC. Yet such His-tag-mediated binding would not lead to channel opening. Likely, this binding is due to additional positively charged residues within the extended cytosolic loop between TM2 and TM3 [63]. This may be necessary because the C-terminus of *E. coli* YidC is much shorter than the typical C-terminal ribosome-binding sites in other bacterial species like *Streptococcus mutants* or *Rhodopirellula baltica* [64,65].

### 3.3. Structural YidC Models

The crystal structure shows YidC as a monomer [16]. Five transmembrane helices form a positively charged hydrophilic groove facing the lipid bilayer. They are arranged in a fashion that does not leave room for a membrane-spanning pore. A subsequent structural model of YidC based on evolutionary co-variation analysis, supplemented by a cryo-electron microscopy reconstruction of a translating YidC–ribosome complex carrying the YidC substrate FoC [48], is in line with this conclusion. The model features a single YidC copy bound to the ribosomal tunnel exit.

The apparent contrast between the pore-free structural models of YidC and our observation of YidC’s channel activity prompted us to interrogate AlphaFold for potential dimer configurations. First, we discarded the resulting antiparallel dimer models since the transmembrane transfer of huge loops is improbable. Moreover, previous research exclusively suggests the existence of parallel dimers [66,67,68]. Second, we selected the parallel dimer with a visible hydrophobic belt, characteristic of membrane proteins (Figure 4A). The resulting model (Figure 4B,C) shows a structure (coordinates in PDB format are provided in the Appendix A) that may expand to form a pore in its center when ligand-activated, e.g., by ribosome binding. Both the N-terminal helix, separated from the rest of the protein by a large periplasmic domain, and the C-terminal helix, exposed to the cytosol and thus offering a potential ribosome binding site, may move outwards. The thus-formed pore would accommodate (i) many of the residues known to interact with nascent chains [69] and (ii) the conserved arginine previously described as essential for YidC functioning [16].

In the predicted parallel YidC dimer, the cytosolic Loop C1 of one protomer is in close contact with the C-terminus of the second protomer (Figure 5A). Previous in vivo crosslinking data combined with mass spectrometry have demonstrated that the surface-exposed C1 loop provides an important docking site for the signal recognition particle (SRP) and also for SecY [12,37]. In addition, a YidC-YidC crosslink was observed when the UV-activatable amino acid derivative para-benzoyl-L-phenylalanine (pBpa) was inserted at position 399 at the tip of the C1 loop [12]. This was verified by expressing either wild-type YidC or YidC (D399pBpa) in *E. coli* BL21 and exposing these cells to UV light to induce a crosslink reaction. Subsequently, after cell breakage, YidC and its crosslinked partner proteins were purified, separated by SDS-PAGE, and analyzed via immune detection using α-YidC antibodies. In YidC (D399pBpa)-expressing cells, three prominent UV-dependent bands at approx. 95 kDa, 110 kDa, and 120 kDa were visible (Figure 5B). These bands have been observed before and were identified as the YidC-SecY crosslink (95 kDa), YidC-Ffh crosslink (110 kDa; Ffh corresponds to the protein component of the bacterial SRP [70]) and the YidC-YidC crosslink (120 kDa) [12]. Some weak UV-dependent bands were also observed for YidC lacking pBpa, which is probably the result of the UV-dependent radical formation of aromatic acids that favor non-specific crosslinks to proteins and nucleic acids [71,72]. In summary, these data demonstrate that the C1 loop of YidC is involved in YidC dimer formation. This was further validated by inserting pBpa at position 544 of the C-terminus of YidC (Figure 5A). After the UV exposure of whole cells, α-YidC antibodies recognized the 120 kDa YidC dimer band (Figure 5C) again. Thus, the observation that both the C1 loop as well as the C-terminus of YidC are involved in YidC dimer formation supports the predicted YidC dimer structure.

### 3.4. Stoichiometry of the Reconstituted YidC

Next, we determined the oligomeric state of YidC in proteoliposomes. Therefore, we fluorescently labeled YidC with Atto-488-maleimide and used FCS to count the number of YidC molecules per liposome, as described previously [56]. In short, FCS determined the number of proteoliposomes per focal volume. Dissolving them with a mild detergent (3% octyl glucoside, OG) increased the number of fluorescent particles three-fold. Each fluorescent particle corresponded to a lipid-detergent-protein micelle (Figure 6A). The subsequent addition of a harsh detergent (1.1% sodium dodecyl sulfate, SDS) doubled the number of particles, indicating that OG micelles contained an average of one YidC dimer. That is, each liposome harbored about three YidC dimers.

To eliminate the possibility that interactions with the detergent affected YidC’s oligomeric state, we performed a blue native PAGE analysis of *E. coli* inverted inner membrane vesicles (IMVs) from the wild-type cells with a native level of YidC expression. In the absence of SDS, Western blot analysis showed an abundance of YidC dimers, which agrees with previous data [30]. In contrast, in SDS, we detected only the monomeric form (Figure 6B). In addition to the strong band at approx. 140 kDa on BN-PAGE, two weaker bands were recognized in INV at approx. 200 kDa and 400 kDa. YidC can form different assemblies with other components of the protein transport machinery [73]. The 200 kDa complex was previously identified as a YidC-SecDF complex [74,75], while the 400 kDa complex likely corresponds to the SecYEG-YidC-SecDF-SurA complex [74]. Our results demonstrate a significant dimerization propensity of YidC at its cellular concentration.

Finally, we performed single-molecule fluorescence experiments to prove that YidC dimers were present in planar lipid bilayers. Since the fluorescence intensity of single fluorophores observable in free-standing planar bilayers is significantly lower than in bilayers near the objective, we performed stepwise bleaching experiments of the labeled YidC complexes in supported and suspended lipid bilayers. The latter model removes the protein from the support because the protein-harboring membrane rests on a layer of streptavidin crystals [76], as schematically shown in Figure 6C. Membrane proteins are free to diffuse in the valleys between streptavidin pillars. To form supported or suspended bilayers, we used empty vesicles and proteoliposomes in a ratio of 1000:1. We analyzed the bleaching of individual fluorescent spots (Figure 6C,D). It appeared that, apart from much larger aggregates, which showed a gradual decrease in fluorescence during bleaching, there were two major populations of particles. Particles in the first population bleached in one step and those in the second in two. As both types of bilayers, supported and suspended, showed similar results, we pooled together the data from both systems for the monomer-dimer distribution (Figure 6E). Thus, a fifth of the observed complexes, corresponding to one third of the total YidC molecules, existed as dimers, with the rest forming monomers.

## 4. Discussion

The insertase YidC mainly facilitates the transfer of single-spanning membrane proteins in a Sec-independent manner. It can also enter the SecYEG and SecDF complexes and form complexes with other membrane proteins, such as PpiD and YfgM [10,37,73,77,78]. YidC’s currently envisioned polypeptide insertion mechanism does not require a pore—in contrast to the operation mode of translocons (e.g., SecYEG). The insertase merely primes the lipid bilayer at its interface for polypeptide insertion. It is unknown how the membrane barrier to small molecules, like ions, is maintained by the primed bilayer or during polypeptide insertion.

In contrast, by exploiting an elaborate gating mechanism, the SecYEG translocon maintains the barrier to small molecules. Its pore is closed in the resting state [35], opens upon binding a signal sequence [79] or other ligands [54,56], and responds to membrane potential [57]. Observing a gated YidC pore raises the question of whether it may serve the same purpose. If it were used for protein insertion, the pore would provide the means for maintaining the barrier to small molecules while allowing the passage of large polypeptides. Observing a dimeric configuration that accommodates many residues interacting with the polypeptide (Figure 4) supports this notion. Additionally, single-channel conductivity indicates that the channel diameter is only 20% smaller than that of SecYEG [35], i.e., the pore would be large enough to accommodate an unfolded polypeptide chain.

Electrophysiological studies on Oxa1 [55], the YidC homolog in the mitochondrial inner membrane 38, further support the pore-forming ability of YidC. Oxa1 was shown to form a cation-selective channel that responds to mitochondrial export signals. However, the Oxa1 channel appears larger than the YidC channel, which could reflect Oxa1′s propensity to form tetramers in the mitochondrial membrane [80,81]. Oxa1 is primarily responsible for inserting the hydrophobic core subunits of the respiratory chain complexes [82]. Its restricted substrate specificity differs from EcYidC, which can insert membrane proteins with varying hydrophobicity and polarity [83].

As we show here, the ribosome also interacts with the C-terminal deletion YidC mutant, meaning that another interaction site in addition to the proposed C-terminus [31] is present, for example, loops between helixes 2–3 and 4–5, as was reported earlier [48,63]. The fact that we observed YidCΔC channel activity in the presence of ribosomes (Figure 3) indicates that there are other elements in addition to the C-terminus responsible for dimerization. The AlphaFold structure suggests two such regions: First, the apparent hydrophobic interactions between helix 1 (residues S3 to K27) of one protomer and the penultimate helix (residues D488 to W508) of the other protomer. Second, the in-parallel helices arranged from both protomers (residues Q387 to G398) at the cytosolic terminus of the potential dimer, contributing to dimer formation via individual polar or more general hydrophobic interactions. In general, it should be noted that specific predictions of the AlphaFold model for interactions with individual amino acids should be taken with a grain of salt. The general prediction for the interaction of larger secondary structure elements or domains are more reliable.

Our BN-PAGE and FCS experiments with YidC vesicles and IMVs showed a predominantly dimeric form both in native *E. coli* membranes and reconstituted proteoliposomes (Figure 6A,B). In comparison, the single-molecule experiments on the suspended and supported bilayers with lower YidC-to-lipid ratios, due to the dilution of YidC required for single-molecule microscopy, showed a predominantly monomeric form (Figure 6C–E). We conclude that YidC can exist in dimeric and monomeric forms in the lipid bilayer, with an equilibrium between the dimeric and monomeric forms.

Significantly, the FCS and photobleaching experiments were conducted in the absence of ribosomes or RNCs, the electrophysiological recordings were carried out in the presence of ribosomes, and the blue native pages and crosslinking results were obtained in the presence of both ribosomes and RNCs. Since all experiments yielded evidence for dimers, the effect of the ligand on dimerization must be weak, if it exists at all.

The dimer resolves the free energy penalty that the charged arginine on the monomer would create when facing the bilayer’s interior. Our observation of monomers coexisting with dimers does not necessarily suggest a functional purpose for the arginine in the monomer. This would only be the case if we were observing the monomer at concentrations typical of living cells. Instead, we detected monomers only in supported bilayers (Figure 6E), where YidC was very diluted. In other words, in our bilayer experiments, YidC was probably present at concentrations below its (as yet unknown) dimerization constant. A similar situation can be observed with model channels such as gramicidin or even aquaporins. We found that the glycerol facilitator (GlpF) can be monomeric when diluted below a certain threshold [84], whereas no one doubts that this aquaporin exists as a tetramer in living cells. The ability of YidC to exist in a monomeric form in addition to a dimeric form upon dilution explains why YidC readily forms complexes with other membrane proteins, e.g., a tetrameric complex with SecYEG [10].

The structural model built by AlphaFold (Figure 4) supports the hypothesis that the dimer is the pore-harboring entity. On the outside, the dimer has a hydrophobic belt that prevents the leakage of small molecules at the protein–lipid interface. At rest, the space between the two monomers is rather small, leaving little room for unspecific leakage. When occupied by a hydrophobic polypeptide, the space between the monomers is increased. However, the overall hydrophobic interior is unlikely to favor the massive partitioning of ions and water. In contrast to SecYEG, YidC does not accommodate hydrophilic polypeptides with multiple charges; hence the sophisticated sealing mechanism of SecYEG, consisting of a plaque and a hydrophobic belt, is superfluous. The observation of eight fusion events with an average amplitude of ~54 pA corroborates the model (Appendix A). This amplitude indicates the presence of roughly three YidC channels per vesicle. At the same time, FCS revealed the presence of three YidC dimers per vesicle (Figure 6A), suggesting that only one channel may be formed per dimer. The conclusion aligns with the previously made observation that a genetically fused YidC dimer is functional even if one monomer is inactivated [85], suggesting that a pore might still be formed in the active/inactive YidC dimer. Our in vivo crosslinking experiments also support the AlphaFold-predicted dimer structure. They show that the C1 loop and the C-terminus of YidC participate in dimer formation (Figure 5).

How a TM would exit the pore formed by a YidC is unknown, but the dimer could of course open at the dimer interface. The conserved arginines should increase the free energy of the hydrophobic segment, thus promoting the segment’s exit into the lipid phase.

Although compelled by the visualization of pore-lining interaction sites with the nascent chain, the projection that only the dimer facilitates polypeptide insertion may not be valid. A cryo-electron microscopy study with a resolution of ~8 Å puts the nascent chain on the surface of a ribosome-bound YidC monomer [48]. Structural data supporting the YidC dimer’s formation suggest that the pore region is formed by transmembrane helixes TM2 and TM3 [31]. However, this structure (resolution 11–14 Å) missed the TM1 earlier shown to interact with the nascent chain [86] and the periplasmic domain. Our AlphaFold prediction suggests that TM1, TM2, and TM3 form the pore.

## 5. Conclusions

YidC is able to form ion-conducting channels when activated by ribosomes. This observation can only be reconciled with the published pore-lacking YidC structures when considering the formation of oligomers. We confirmed dimer formation using a whole arsenal of experimental methods: fluorescence correlation spectroscopy, single molecule imaging, site-directed crosslinking and blue native pages. In addition, we used the advantages provided by artificial intelligence (AlphaFold) to show that dimer formation is plausible and how the two YidC subunits may be positioned relative to each other. Future research will be focused on obtaining evidence that polypeptides are indeed partitioning from the dimer into the lipid membrane.

## Figures and Tables

**Figure 1 biomolecules-13-01774-f001:**
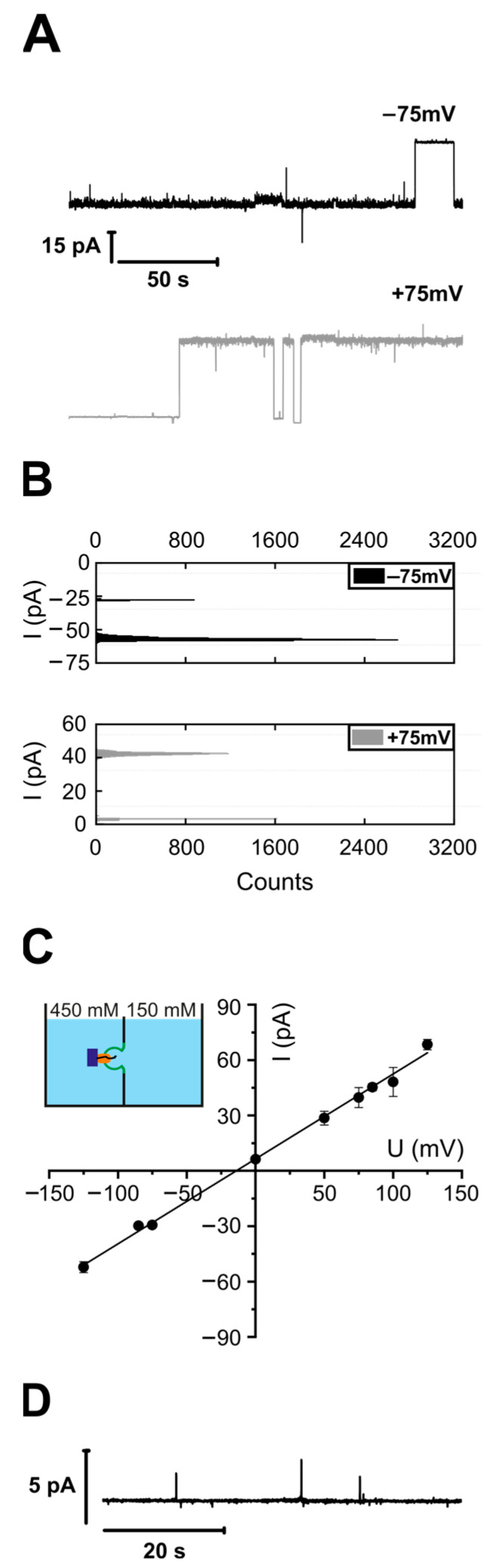
YidC exhibits channel activity after FoC-RNC binding. (**A**) Current traces show single channel openings and closings recorded at different voltages. The YidC-RNC complex was reconstituted into the planar bilayer via a vesicle fusion assay conducted in asymmetric salt conditions: 450 mM KCl in the *cis* compartment, 150 mM KCl in the *trans* compartment, and 50 mM K-HEPES, pH 7.5, in both compartments. YidC proteoliposomes were added to the chamber with a pre-formed lipid bilayer from the hypertonic side. The open states are noisier than the closed states due to the frequent flickering of the pore. (**B**) Current histograms corresponding to each trace in (**A**). The distance between each of the two neighbor peaks on the histogram corresponds to the current through a single channel. (**C**) The current–voltage plot of the YidC-RNC complex was obtained from single channel amplitudes at different voltages (dots) from traces, like in (**B**). The linear fit yielded the single channel conductivity g = 459 ± 20 pS and a reversal potential of −9 mV resulting from the slight asymmetry in the conductance for cations and anions. A colored scheme shows the chamber, with two compartments filled with solutions of the indicated ionic strength. The compartments are separated by the lipid bilayer (green), containing YidC (orange) and bound RNC (ribosome as a purple rectangle with a nascent chain as a black line). (**D**) Negative control: lack of channel activity in the absence of FoC-RNC. At least three biological replicates were used for electrophysiological measurements.

**Figure 2 biomolecules-13-01774-f002:**
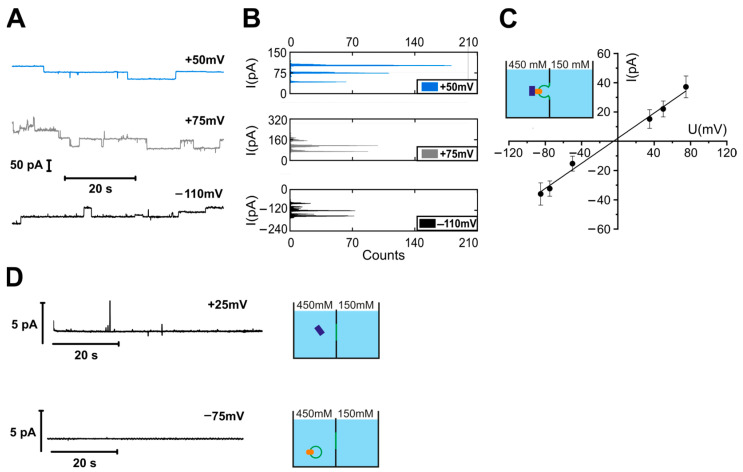
YidC exhibits channel activity after ribosome binding. (**A**) Current traces show single channel openings and closings recorded at different voltages. (**B**) Current histograms corresponding to each trace in (**A**). The distance between each of the two neighbor peaks on the histogram corresponds to the current through a single channel. (**C**) The current–voltage plot of the YidC–ribosome complex was obtained from single channel amplitudes at different voltages (dots) from traces, like in (**B**). The linear fit yielded the single channel conductivity g = 436 ± 20 pS. The reversal potential of −5 mV indicates the lower asymmetry for cations and anions’ permeabilities, compared to Figure 1. The YidC–ribosome complex was reconstituted into the planar bilayer by vesicle fusion, as in Figure 1. (**D**) Negative controls: lack of channel activity without ribosomes (lower trace) or YidC (upper trace).

**Figure 3 biomolecules-13-01774-f003:**
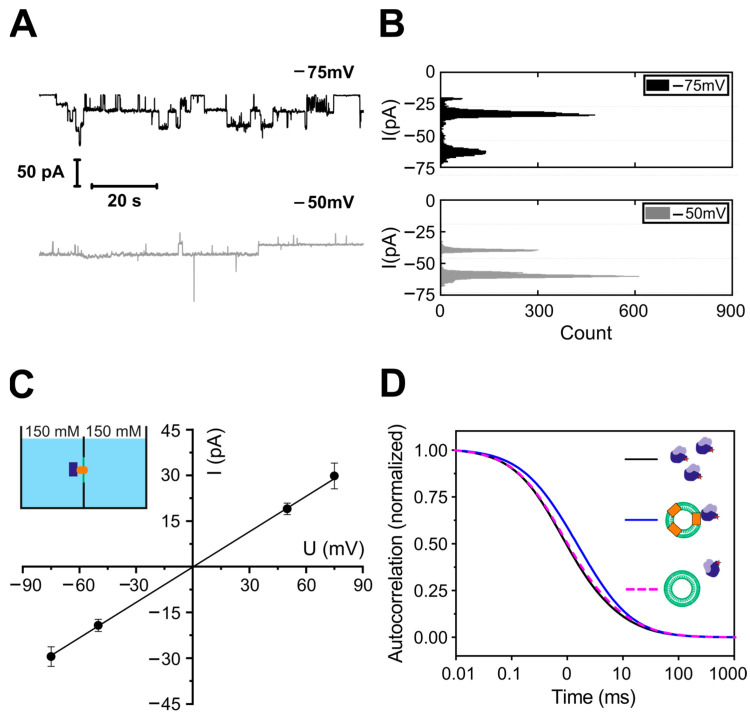
Ribosomes activate YidCΔC. (**A**) Channel activity of the planar bilayers with the reconstituted YidCΔC after adding ribosomes. The experiments are analogous to the ones in Appendix A. (**B**) Histograms of the recordings shown in (**A**). (**C**) Single channel amplitudes (dots) and the current–voltage characteristic of the YidCΔC–ribosome complex (line) at different voltages obtained from traces, like in (**A**). Single channel conductivity g = 392 ± 30 pS. (**D**) The binding of ribosomes to YidCΔC vesicles observed via fluorescence correlation spectroscopy. Autocorrelation curves of labeled ribosomes in solution (black), of ribosomes mixed with YidCΔC proteoliposomes (blue), and ribosomes mixed with empty vesicles containing a lipid label (pink dashed line). The ribosomes and vesicles have comparable diffusion times across the confocal volume, whereas their complex diffuses slower. No binding could be observed when ribosomes were mixed with empty vesicles. Experiments were conducted in 150 mM KCl and 50 mM K-HEPES, pH 7.5. At least three biological replicates were used for the FCS measurements.

**Figure 4 biomolecules-13-01774-f004:**
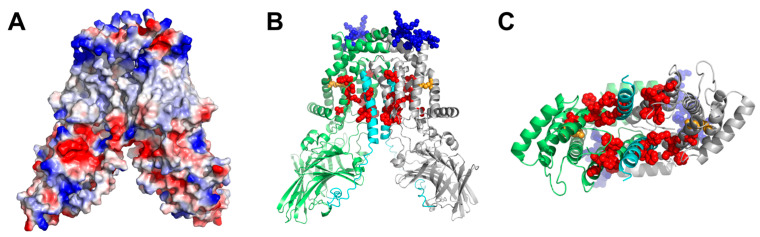
Model of the YidC dimer created with AlphaFold. (**A**) Surface charge representation of the parallel dimer model. The two closely interacting alpha-helical domains form a consistent hydrophobic belt. Color code: blue—positive charge, red—negative charge, white—no charge. (**B**) Cartoon representation of the parallel dimer model (top: cytosol, bottom: periplasm). The C-terminal amino acids (blue spheres) are exposed and thus available for potential interaction with a ribosome. The highly flexible N-terminus (cyan) was modeled to participate in the parallel helix formation and might contribute to the hydrophobic belt and substrate translocation or gating. Amino acids previously described as involved in substrate interactions are highlighted with red [69] and orange (R366) [16] spheres, respectively. (**C**) Model viewed from below and with the beta domains removed. Notably, most of the amino acids described as potential interactors with YidC substrates are grouped in the center of the parallel complex, many of them even facing the inside. The model was analyzed and visualized with PyMOL.

**Figure 5 biomolecules-13-01774-f005:**
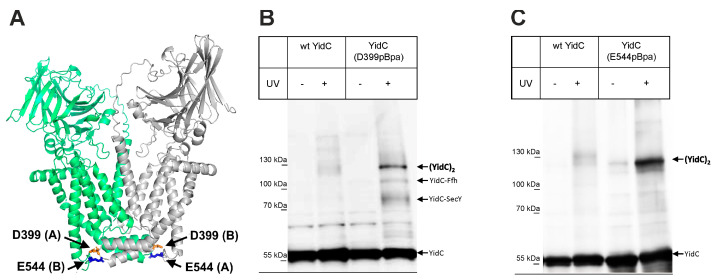
The C1 loop and the C-terminus of YidC are involved in YidC dimerization. (**A**) Location of residues D399 within the C1 loop and E544 at the C-termini in the parallel YidC dimer model, which were replaced by para-benzoyl-L-phenylalanine (pBpa). (**B**) In vivo photo-crosslinking performed with *E. coli* BL21 cells expressing *yidC* from the plasmid P*_lac_secYEG-yidC*. WT refers to wild-type YidC and D399pBpa to a YidC variant with pBpa inserted at position 399. After UV exposure on ice and cell breakage, YidC and its crosslinked partner proteins were separated by SDS-PAGE and, after western transfer, decorated with α-YidC antibodies. A sample without UV exposure served as a control. The crosslinks are indicated and have previously been confirmed via mass spectrometry [12]. (**C**) As in B, pBpa was inserted at position 544 within the C-terminus of YidC. YidC and its crosslinked dimer are indicated. The crosslink experiments were performed several times (n ≥ 3) as biological replicates, and a representative gel is shown.

**Figure 6 biomolecules-13-01774-f006:**
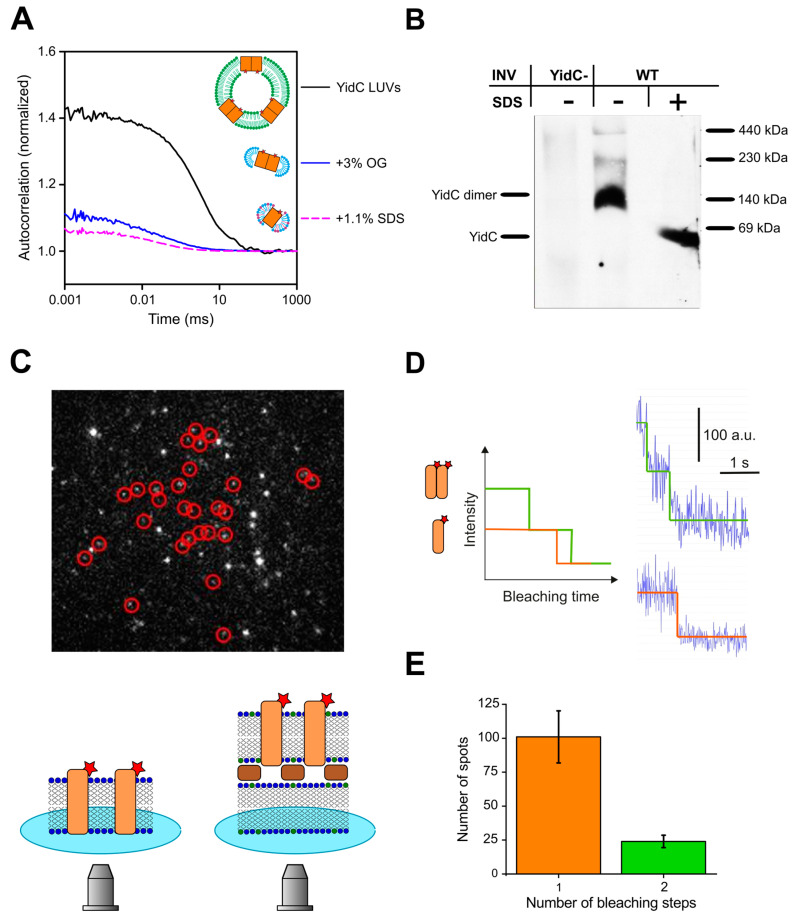
Stoichiometry of YidC. (**A**) FCS served to obtain the YidC stoichiometry in reconstituted liposomes. The number per confocal volume of YidC vesicles, YidC-OG-lipid, and YidC-OG-lipid-SDS micelles was derived from the amplitude A of the corresponding autocorrelation curve as 1/(A − 1). In the same order, the number of resulting particles amounted to about 2.5, 7.5, and 15, respectively, meaning that the vesicles contained, on average, three YidC dimers. The color code for cartoon schemes is the same as in Figure 3, with OG molecules depicted in cyan and SDS molecules in dark red. The red stars indicate the fluorescent label. (**B**) YidC forms dimers in its native environment. INVs of wild-type *E. coli* cells (wt) or a conditional YidC depletion strain (YidC-) were analyzed by BN-PAGE and, in the course of Western blotting, decorated with an antibody against YidC. The BN page clearly shows a YidC dimer at 140 kDa (lane 2), which can be dissolved to a monomeric state via SDS treatment before loading the BN-PAGE (lane 3). (**C**) An exemplary single molecule fluorescence bleaching experiment. Such experiments were conducted on supported (lower scheme on the left) and suspended (lower scheme on the right) lipid bilayers, which were formed from YidC vesicles mixed with empty vesicles (1:1000). The color code of the schemes is the same as above, with streptavidin crystal shown as brown rectangles, biotinylated lipid shown in green, and cover slide glass in blue. The spots marked by red circles were later analyzed. The brighter spots correspond to aggregates. (**D**) Spots in the red circles showed one- and two-step bleaching as schematically shown on the left, with exemplary intensity traces on the right. (**E**) The histogram of the number of bleaching steps generated after analyzing the spots from different experiments shows that 101 spots were bleached in one step and 24 in two steps. That is, approx. 1/5 of the spots corresponded to YidC dimers. At least three biological replicates were used in each experimental approach.

## Data Availability

Available from the corresponding author upon request.

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
