# Peer review of "YidC from Escherichia coli Forms an Ion-Conducting Pore upon Activation by Ribosomes"

_biomolecules, 2023, doi:10.3390/biom13121774_

Round 1

Reviewer 1 Report

Comments and Suggestions for Authors

The paper from Knyazev et al. presents new data on the membrane insertase YidC, which assists with the insertion of membrane proteins. The authors find channel activity upon interaction with ribosomes as well as ribosome-nascent chain complexes. They rationalize these results in the light of a dimeric model of YidC.

This is an interesting study that leaves a lot of questions. I trust the authors' data, although what this says about the functional oligomeric state remains unclear. Indeed, I think even the authors are unsure given this line at the end: "...the projection that only the dimer facilitates polypeptide insertion may not be valid". Nonetheless, it's good to occasionally question existing models, particularly when presented with new data that will ultimately need to be reconciled regardless of whether the monomer or dimer (or both!) is the functional form.

It's noted in the introduction that an exposed arginine is unusual in a membrane protein, which suggests to me that it probably has a functional purpose. Placing it at the center of the dimer would resolve the free-energy penalty *but* the monomeric form of the protein is still found, again suggesting a functional purpose for the retained arginine, such as locally distorting the membrane to make insertion easier. Can the authors comment on this?

How might the dimer prevent leakage of small molecules during translocation?  Does the modeled structure reveal anything akin to the hydrophobic pore ring of SecY?

Also, how would a TM exit the dimer pore?

Finally, I didn't understand this in the discussion:

"At the same time, FCS revealed the presence of three YidC dimers per vesicle (Fig. 6A), suggesting that only one channel may be formed per dimer. The conclusion aligns with the previously made observation that, in a genetically fused YidC dimer, each monomer can act independently of the other [71]."

If the dimer is the functional unit, I would expect one channel per dimer, with both monomers contributing equally. But the result of [71] was that inactivating one of two fused monomers still permitted function. So how do these two sentences agree with one another? 

Reviewer 2 Report

Comments and Suggestions for Authors

In this manuscript, the authors described their study on the ability of YidC in E. coli to insert proteins to membranes by forming a ion-conductive pore. It is argued by the authors that the pore is formed as a YidC dimer. The authors also showed that YidC can be activated by both the natural substrate and ribosomes. Overall this is an interesting topic and the authors have got some good preliminary data. Before considering the manuscript for publication, I have some several questions that I hope the authors can clarify, as well as some comments/suggestions for the authors.

1. I am a little confused about the the observation that ribosomes could activate the YidC C-terminal deletion mutant. The structural data suggests that C-terminal is likely required for dimerization. Does this mean that ribosome could activate the YidC monomer? This is an interesting observation and it would be great if the authors can look into this in more detail.

2. The two main pieces of data: activation of YidC and dimer confirmation, I'm having a hard time connecting this 2 pieces together. Could the author design some experiment to directly show the activity of YidC as a dimer? This will greatly add to the scientific significance of this paper. 

3. I would suggest the authors to introduce FCS the first time it was mentioned (Fig 3C) instead of later.

4. In Fig 3, the current histograms were not shown. Legends on 3B needs to be revised.

In summary, the manuscript can be considered for publication, but some revision and clarification is required. 

Comments on the Quality of English Language

The Quality of English is generally good of this manuscript.

Round 2

Reviewer 2 Report

Comments and Suggestions for Authors

The authors addressed the reviewer's comments appropriately and now the manuscript is much improved. I'm happy to recommend this revised version for acceptance.